# The Backbone of Success of P,N-Hybrid Ligands: Some Recent Developments

**DOI:** 10.3390/molecules27196293

**Published:** 2022-09-23

**Authors:** Martin B. Smith

**Affiliations:** Department of Chemistry, Loughborough University, Loughborough, Leics LE11 3TU, UK; m.b.smith@lboro.ac.uk

**Keywords:** amine groups, chelate effect, coordination chemistry, NMR spectroscopy, P ligands, synthesis

## Abstract

Organophosphorus ligands are an invaluable family of compounds that continue to underpin important roles in disciplines such as coordination chemistry and catalysis. Their success can routinely be traced back to facile tuneability thus enabling a high degree of control over, for example, electronic and steric properties. Diphosphines, phosphorus compounds bearing two separated P^III^ donor atoms, are also highly valued and impart their own unique features, for example excellent chelating properties upon metal complexation. In many classical ligands of this type, the backbone connectivity has been based on all carbon spacers only but there is growing interest in embedding other donor atoms such as additional nitrogen (–NH–, –NR–) sites. This review will collate some important examples of ligands in this field, illustrate their role as ligands in coordination chemistry and highlight some of their reactivities and applications. It will be shown that incorporation of a nitrogen-based group can impart unusual reactivities and important catalytic applications.

## 1. Introduction

Phosphorus compounds are of widespread fascination due to their importance in organic, coordination/organometallic chemistry, catalysis, and numerous other applications. Many such P-based compounds are derived from simple fundamental building blocks such as white phosphorus (P_4_) for example. Various industrial processes for accessing simple P-based compounds are well known and, nowadays, current challenges in generating such compounds more efficiently [1] and sustainably are being actively pursued [2]. Interest has spurred in converting P_4_ into useful compounds directly [3,4] and this area will no doubt continue to be a highly important area going forward.

The following review will provide a brief update, from the Author’s perspective, of selected examples of chelating diphosphines with a central nitrogen functional group in the backbone and illustrate the diverse behaviour(s) imparted by this additional donor atom. The focus will be on ligand design and synthesis protocols, ^31^P{^1^H} NMR spectroscopy as a useful tool for assessing purity and characterisation and an illustration of how such ligands are used, primarily in coordination chemistry and catalysis. This review is by no means exhaustive but will highlight the growing emergence of these ligands versus their all-carbon backbone counterparts which historically have been known for several decades. Depending on the diphosphine ligand, 4-, 5-, 6- (Figure 1) and larger chelate/macrocyclic rings with one (or more) –NH–/–NR– sites are formed upon complexation to a range of typical transition metals. The present review aims to embrace the Reader with a perspective of the importance of these ligands and their continuing prominence going forward. These ligands constitute important families to the already extensive number of known phosphorus compounds. As will be illustrated, the tuneability of phosphines thereby precisely controlling properties such as electronic effects, sterics, asymmetry, solubility, chirality etc is applicable to the diphosphines discussed here.

Unlike general synthetic phosphorus methodologies widely used for constructing P–C bonds [5], the transformations shown in Figure 1 are simple condensations and employ, typically, secondary chlorophosphines or secondary phosphines as key starting reagents. In many instances, especially the Phospha-Mannich based reactions [6,7], work-up and isolations are straightforward and target P(III) compounds can often be accessed in good to high yields. Furthermore, in an industrial context, these reactions are important in the industrial preparation of flame retardants for cotton based materials using [P(CH_2_OH)_4_]^+^ salts and condensation with urea [8]. The utility of this approach will be highlighted, using suitable examples, in the following sections.

## 2. P–N Chemistry

### 2.1. Synthesis of Selected Examples

The ligand **1a** (Figure 2), bearing a phenyl substituent (X = Y = H), has been known for over 25 years [9] and related analogues have been described [10]. The pyridyl analogue, **1b**, was first reported in 2000 by Woollins and co-workers and prepared from 2-aminopyridine, NEt_3_, and ClPPh_2_ at 0 °C and isolated as a white powder in 78% yield [11]. Reaction of an in situ generated lithiated amide, with 1 equiv. of Ph_2_PCl, afforded air stable **1c** as a white solid in 66% yield and showed a ^31^P signal at 38.5 ppm [12]. A single crystal X-ray structure analysis of **1c** revealed a P–N distance of 1.702(13) Å. Ligand **1c** could function in a P-monodentate fashion, but also through one of the phenyl rings on the benzhydryl group as a tethered η^6^-arene when bound at a Ru(II) piano-stool centre. Pope and co-workers [13] prepared a range of fluorescent phosphinous amides **1d–f** from a stoichiometric reaction of the appropriate primary amine:ClPPh_2_ in CH_2_Cl_2_, and NEt_3_, at 0 °C. It was found that **1d–f** could be isolated as colourless, yellow oils or as a dark orange solid and showed ^31^P signals at 42.2, 42.1, and 26.1 ppm, respectively. These ligands were coordinated to gold(I) and displayed interesting photophysical properties. In 2022, unlike conventional pathways to phosphinoamines (Route A, Figure 1) previously highlighted, an electrochemical route has been reported as an alternative strategy for generating P–N bonds. For example, **1g** could be accessed through coupling of secondary amines/phosphines and concomitant H_2_ evolution [14]. Utilising a single P–N bond forming step has successfully enabled the preparation of PN_2_-tridentate ligands such as **1h** and **1i** [15,16] which have subsequently been widely studied in various catalytic transformations via metal ligand cooperative catalysis. The synthesis of **1h** was accomplished using NEt_3_/^n^BuLi, ^t^Bu_2_PCl, and 6-[(diethylamino)methyl]pyridine-2-amine. The tridentate phosphinoamine, **1j**, was prepared from Cy_2_PCl and BzNH_2_ in C_7_H_8_, in the presence of excess NEt_3_, and was isolated as a colourless solid in 69% yield [17]. The ^31^P NMR of **1j** showed one resonance for the PN atom at 32.6 ppm (as a dd) and −12.6 ppm (as a d for the -PCy_2_ groups, Table 1). Other multidentate ligands, such as **1k** [18], were synthesised by reaction of (2-C_5_H_4_N)CH_2_N(CH_2_CH_2_NH_2_)_2_ with 2 equiv. of ^i^Pr_2_PCl in THF in the presence of excess NEt_3_ to afford the desired product as a pale yellow oil in 88% yield and a ^31^P NMR at 63.9 ppm (in C_6_D_6_). P–N formation can be accomplished, through condensation, upon reaction of 2-aminopyridine and [ClP(μ-N^t^Bu)]_2_ in the presence of excess NEt_3_, in THF, to afford **1l** as a colourless crystalline solid in 32% yield. The ^31^P NMR spectrum showed a signal at 106.4 ppm whilst single crystal X-ray analysis revealed an intramolecular N–H^…^N H-bond spanning the P_2_N_2_ four-membered core [19]. The tripodal phosphinoamine **1m** [20,21] was readily prepared from N(CH_2_CH_2_NH_2_)_3_ and ^t^Bu_2_PCl (3 equivs) in THF in the presence of DBU and isolated as a colourless oil in 95% yield and with a ^31^P NMR signal at 77.9 ppm (in C_6_D_6_). The three NH sites could further be readily deprotonated, with ^n^BuLi and quenched with GaCl_3_, to form a corresponding Ga compound. The ligand **1n** (abbreviated as TPAP) was synthesised from reaction of tris[2-[N-(2-pyridinemethyl)-amino]ethyl]amine and ClP(NMe_2_)_2_, which upon subsequent deprotonation with excess ^t^BuOK in THF gave **1n** as a white solid which displayed a ^31^P resonance at 126 ppm [22,23].

A well-known transformation of tertiary phosphines is their oxidation and this reactivity is also the case for phosphinoamines which react with H_2_O_2_, elemental sulfur or selenium or even BH_3_.SMe_2_ to give the corresponding P(V) analogues. These aminophosphine chalcogenides display a rich and varied chemistry towards a range of alkali metal complexes, frequently involving deprotonation of the NH group [24].

### 2.2. Importance of –N(H)– Backbone Functionality in P–N Ligands

The impact of an –N(H)– group in catalysis can be nicely illustrated by the methyl substituted derivative Ph_2_PN(Me)(2-C_5_H_4_N) **1o** which could be prepared from 2-(methylamino)pyridine in Et_2_O and ^n^BuLi as base followed by quenching with ClPPh_2_ [25]. After workup, **1o** was isolated as a pale yellow crystalline solid in 65%. The palladium(II) catalysed methoxycarbonylation of olefins using **1b** as ligand, bearing an –NH– spacer, was found to be an efficient catalyst system. In contrast, when using **1o** instead, a significantly reduced catalytic activity was observed thereby demonstrating an important effect of the –NH– group. The free –NH– group in these singly substituted phosphinoamines also enables further P–N bond coupling reactions to occur leading to, for example, unsymmetrical P–N–P diphosphinoamines, as will be discussed further in Section 3.2.

Coordinated aminophosphines, bearing an –NH– group, often display excellent hydrogen bonding capabilities and are often investigated using single crystal X-ray crystallography. For example, cis-dichoroplatinum(II) complexes of ester functionalised analogues of **1a** show intramolecular N–H^…^ClPt H-bonding [10]. With the inclusion of an additional acceptor, such as that in **1b**, intermolecular N–H^…^N hydrogen bonding linking two molecules into a dimer pair was observed [11]. Other H-bonding motifs have also been observed but are not discussed here.

Deprotonation of the secondary amine group can also enable amido based ligands to be prepared as illustrated by the elegant work of Velian and co-workers [26]. This group showed that the coordinated Ph_2_PN(H){C_6_H_4_(4-Me)} ligand in a cobalt selenide cluster could be deprotonated, using ^n^BuLi and reacted with FeCl_2_, to form a mixed Co/Fe cluster in which [Ph_2_PN{C_6_H_4_(4-Me)}]^−^ bridges an Fe and Co metal centre. Likewise, a copper complex of **1j**, was found to undergo deprotonation with KH to form a binuclear amido complex in which **1j** functions as a P_3_N-tetradentate ligand [17]. Deprotonation of **1m** with ^n^BuLi was shown to afford trianionic trisamido ligands which could be used to form an interesting array of heterometallic complexes [18,19]. The –NH– group in complexes of pincer ligands (e.g., **1h**, **1i**) play an important catalytic role through metal-ligand cooperation involving dearomatisation/rearomatisation via deprotonation/reprotonation steps [16].

## 3. P–N–P Chemistry

### 3.1. Synthesis of Selected Examples

Small bite-angle diphosphines such as the ubiquitous dppm [bis(diphenylphosphino)methane] have attracted much interest over the years as an excellent chelating or bridging ligand for mononuclear and polynuclear metal centres. Closely related to dppm, namely dppa [bis(diphenylphosphino)amine, Figure 3], has also been widely studied as an excellent coordinating ligand [27,28,29,30]. One further attractive advantage of dppa, over dppm, is the ease by which it is possible to further functionalise the central amine with a range of different substituents and some highlights to demonstrate this behaviour will be reviewed here. The diphosphine, dppa, can readily be synthesised from ClPPh_2_ and (Me_3_Si)_2_NH and isolated as a white solid [31].

One of the most spectacular successes of bis(phosphino)amines in catalysis has focused on extensive studies of PNP ligands that have been reported, in relation to the formation of 1-octene via a Cr catalysed ethylene tetramerisation [32]. A typical PNP ligand is based on Ph_2_PN(^i^Pr)PPh_2_ (**2a**) and an extensive library of other –N(alkyl)– based systems have been reported [33,34]. Suntharalingam and co-workers [35] prepared a simple alkyl chain (diphosphino)amine, Ph_2_PN(C_6_H_13_)PPh_2_ (**2b**) from n-hexylamine and 2 equiv. of ClPPh_2_ in CH_2_Cl_2_ in the presence of NEt_3_ and was isolated as a white solid in 81% yield. A diagnostic ^31^P resonance at δ 62.1 ppm (in CDCl_3_) was observed (Table 1). Homoleptic Group 10 metal complexes were prepared, as their tetrafluoroborate salts, and the Pd/Pt complexes were shown to be extremely potent complexes for CSC mammosphere activity. Additional functionalities have been incorporated into the –N(R)– backbone (**2c**, **2d**) and, in these cases, no further donor atom participation was found in the resulting complexes that were studied [36,37]. Khan and co-workers have used some simple, –N(aryl)– modified, (diphosphino)amines to investigate a range of Au(I) complexes and their aurophilic behaviour [38]. Two ligands studied in this work were Ph_2_PN(Ph)PPh_2_ and 2,6-Me_2_C_6_H_3_N(PPh_2_)_2_ (**2e**) which both react with AuCl(SMe_2_) to afford P-monodentate or P,P-bridging gold(I) complexes. From X-ray diffraction studies of these complexes, strong intramolecular Au^…^Au contacts were observed and displayed excellent luminescent properties with high quantum yields. The same group [39] also used bis(phosphino)amines, here 2,6-^i^Pr_2_C_6_H_3_N(PPh_2_)_2_ (**2f**). for synthesising further Au(I) complexes. An alternative illustration of routes to the synthesis of PNP ligands was provided by Pal and co-workers who prepared Ph_2_PN(2,6-^i^Pr_2_C_6_H_3_)PPh_2_ (**2f**), in a two-step process, from the primary amine/^n^BuLi/ClPPh_2_ in Et_2_O, to give the desired PNP ligand as a yellow solid in 87% yield. The ^31^P NMR spectrum showed a singlet at −6.4 ppm significantly upfield from other –N(arene)– diphosphines of this type. In addition to **2e**, (p-Me)C_6_H_4_N(PPh_2_)_2_ could be used to prepare Cu(I) complexes that exhibited mechanochromic as well as thermochromic luminescent behaviour. This may be attributed to the shortened Cu^…^Cu contacts [40]. We showed that PNP ligands, with ester groups on the central –N(arene)– core (e.g., **2g**) could be synthesised from the amine/ClPPh_2_/NEt_3_ in Et_2_O [41]. Ligands such as **2h** could be prepared from Me_3_SiN(R`)SiMe_3_/2 equiv. PCl_2_R and, depending on the R groups employed, could be obtained as a mixture of *R,S*/*S,S* and *R,R* stereoisomers in different amounts [42].

### 3.2. Importance of –N(R)– Backbone Functionality in P–N–P Ligands

The –NH– backbone functionality in P–N–P ligands is a good donor, for H-bonding, and has been shown to H-bond to various solvents (e.g., dioxane, MeOH, acetone) and anions (e.g., PF_6_^−^) [27,28,29,30].

Furthermore, like dppm, it is possible to deprotonate the –NH– proton to give the corresponding diphenylphosphinoamido anion which could also coordinate to metal centres. To illustrate this point, Kemp and co-workers [43] prepared a five-co-ordinate indium(III) complex In(^i^Pr_2_PNP^i^Pr_2_)_2_Cl by reaction of (^i^Pr_2_P)_2_NLi and InCl_3_ in Et_2_O/THF.

Routes to increasing donor functionality in PNP ligand systems have been realised through the works of several groups. Balakrisna and co-workers [44] described a new terpyridine based diphosphinoamine **2i** (Figure 4) prepared from reaction of the parent amine and ClPPh_2_, in CH_2_Cl_2_ and the presence of NEt_3_. The phosphine **2i** was isolated as a colourless solid in 48% yield and displayed a characteristic ^31^P resonance at δ 69.3 ppm (in CDCl_3_). From a single crystal X-ray analysis, the two P–N bond distances are 1.713(2) and 1.721(2) Å with a P–N–P angle of 114.46(12)°. The sum of the angles around the nitrogen centre is 359.77° indicating a planar geometry. Diphosphine **2i** was shown to exhibit various ligating motifs (P,P-chelate, P,P-bridging, or as a multidentate system using all P_2_N_3_ donor atoms). Braunstein and co-workers [45,46] prepared a series of bis(phosphino)amines, including the thioether ligands **2j** and **2k** [R = PhCH_2_, CH_3_(CH_2_)_5_]. These were prepared from the corresponding primary amines and ClPPh_2_, in the presence of NEt_3_, in Et_2_O at 0 °C. Both **2j** and **2k** showed a single peak at δ 62.9 ppm in their ^31^P{^1^H} NMR spectra and both ligands could be used to afford heterodinuclear and trinuclear metal complexes through P_2_S-binding. Roodt and co-workers [47] developed new pathways to potentially water-soluble P–N–P ligands such as **2l**. Hence, the N-Boc ligand was synthesised from the free primary amine and ClPPh_2_, in CH_2_Cl_2_, in the presence of NEt_3_. The product was isolated as a white solid in 81% yield and the ^31^P showed a signal at 63.8 ppm (in CD_2_Cl_2_). This ligand could be “protected” by coordination to an {Re(CO)_3_} fragment, whereupon the protecting group was removed with CF_3_CO_2_H followed by neutralisation to give a free pendant amine site.

Unsymmetrical R`_2_PN(R)PR``_2_ ligands [48,49,50] could also be synthesised through stoichiometric reaction of two different chlorophosphines and MeNH_2_ or PhobPN(H)Me (Phob = phobane) and Ar_2_PCl in the presence of NEt_3_. There was also a strong correlation with iminobisphosphine PPN products formed under certain experimental conditions [49,50,51]. On this theme, another intriguing reaction to note, for bis(phosphino)amines, is their rearrangement to iminobisphosphines, which is reversible and promoted by protonation/deprotonation [52]. The N=P–P group is isomeric to the P–N–P systems. The bis(phosphino)amine **2m** reacts with HBF_4_ in CH_2_Cl_2_ to give **2n**, as the tetrafluoroborate salt in quantitative yield. Deprotonation of **2n** with NEt_3_ regenerated **2m**. Agapie and co-workers have also shown the importance of aluminum induced isomerisation of PNP to PPN ligands and relevance to Cr based ethylene tetramerisation catalysis [53].

The impact of an –N– substituent is clear with regard to the relationship between steric effects and 1-octene/1-hexene selectivities using PNP/Cr complexes. Roodt [54] introduced a steric parameter to describe the steric bulk at the N atom of a range of bis(phosphino)amine ligands, a parameter similar to the Tollman cone angle widely appreciated. Furthermore, simple –N(R)– group manipulation (R = H, Me, Et) of bis(diphenylphosphino)amines can afford a range of novel, unique, polynuclear gold(I) sulfido complexes as reported in 2020 by Yam and co-workers [28].

Finally, our group has shown [41] that the P–N bond undergoes room temperature methanolysis in which one bond is cleaved affording a phosphinoamine, bearing an –NH– group, and a cis MeOPPh_2_ phosphinite bound at a Pt(II) square planar metal. Furthermore, C–H activation of an *ortho* C–H_arom_ was found to occur, within the coordination sphere, affording a five-membered metallacycle that was confirmed by single crystal X-ray crystallography.

## 4. Bicyclic P–N Chemistry

Radosevich and co-workers [55] reported the synthesis of two phosphorus triamides **3a/3b** (R = Me, ^i^Pr, Figure 5) from the reaction of triamines and PCl_3_ and NEt_3_ in a mixed THF/Et_2_O solvent to give **3a** in 80% yield as an off-white solid. The N-propyl analogue, **3b**, was prepared in 83% yield. For **3a** the observed ^31^P shift was at 159.8 ppm and the P–N distances were found to be 1.7014(14) Å and 1.7190(13) Å whereas the pseudoaxial nitrogen was longer [1.7610(12) Å]. The synthesis of an unusual 10-aza-9-phosphatriptycene, [56] was achieved by reaction of the brominated tertiary amine and ^t^BuLi, followed by addition of (ArO)_3_P (1 equiv) to give **3c** in 71% yield and whose ^31^P displayed a signal at −77.0 ppm.

## 5. P–C–N Chemistry

### 5.1. Synthesis of Selected Examples

Careful monitoring of the reaction conditions for the preparation of the ^t^Bu phosphine **4a** (Figure 6), as a yellow oil, were required and involved performing the reaction at r.t. in CH_2_Cl_2_ with 1.5 equiv. of amine and Ph_2_PCH_2_OH [57]. The ^31^P NMR shift (in CH_2_Cl_2_) was −16.7 ppm (Table 1). Under dynamic vacuum, the reaction of aniline and Ph_2_PCH_2_OH gave the phenyl analogue, **4b**, in 95% yield and showed a signal at −19.4 ppm (in C_6_D_6_). The pyrimidine ligand **4c** was prepared from Ph_2_PH, (CH_2_O)_n_ and 2-aminopyrimidine, in C_7_H_8_ and isolated in 93% as a white solid [58]. The ^31^P chemical shift was indicative of single substitution. The 1,10-phenanthroline functionalise ligand, **4d**, could be isolated in 63% yield as a yellow solid, from the in situ reaction of HPPh_2_, (CH_2_O)_n_ and 2-amino-1,10-phenanthroline [59]. The PN_2_-tridentate behaviour could be demonstrated through coordination to Co^2+^ and Ni^2+^ affording octahedral complexes with two ligands per metal. Here, the ligands are either PN_2_- or N_2_-coordinated. In 2021, Ren and co-workers showed that **4d** could afford a trinuclear Au_2_Ag complex through PN_2_-coordination [60]. An alternative entry route to Ph_2_PCH_2_OH could be achieved through base (NEt_3_) treatment of [PPh_2_(CH_2_OH)_2_]Cl [61]. Accordingly, this was used to access **4e** in 95% yield and which displayed a typical ^31^P resonance at −35.9 ppm. The corresponding oxide was obtained through reaction with aq. H_2_O_2_ in CHCl_3_. The phosophino-aza crown **4f** could be prepared from the parent amine, HPPh_2_, CH_2_O in THF at 60 °C as a colourless oil in 80% isolated yield and showed a ^31^P chemical shift at −26.3 ppm [62]. Using a Re(I) complex of **4f** incorporating the macrocyclic ring promoted binding of Group 2 metal ions. In 2021, our group recently described an unusual approach for obtaining coordinated P–C–N ligands from a bridging P–C–N–C–P ligand that was promoted by internal acid protonation from an arene group located on the central N atom [63]. This was shown to be a remarkably clean reaction affording **4g** [and only RuCl_2_(p-cym)(Ph_2_PH), cym = cymene] and the progress of the reaction could be carefully monitored, in solution, by NMR spectroscopy. In the absence of a metal, this reaction did not proceed cleanly. It should also be added that P–C–N ligands can also undergo P–C cleavage affording a secondary phosphine complex. Reaction of tris(2-aminophenyl)amine in CH_2_Cl_2_ with Ph_2_PCH_2_OH (3 equiv.) in the presence of CaH_2_ to remove H_2_O, gave the phenyl **4h** compound as a pale white solid in 82% yield and with a ^31^P of −19.6 ppm (in C_6_D_6_) [64]. A similar approach could be employed, for the isopropyl analogue of **4h**, using ^i^Pr_2_PCH_2_OH affording a white powder in 90% yield [δ(P) 3.1 ppm, C_6_D_6_] [65]. Finally, the success of Phospha-Mannich condensations, with Ph_2_PCH_2_OH, could be realised for the synthesis of highly decorated dendrimers with multiple terminal N–C–PPh_2_ functionalities [66].

The synthesis of tripodal aminomethylphosphines, **4i** [67], **4j** [68,69,70,71,72,73] and **4k** [74] have successfully been achieved through threefold condensation using P(CH_2_OH)_3_ and three equiv. of the appropriate amine, either in dynamic vacuum or using toluene as solvent in an azeotropic distillation to remove water. For **4k**, the use of [P(CH_2_OH)_4_]Cl as a P(V) precursor and reaction with 4 equiv. of amine afforded the corresponding tetraaminoalkylphosphonium salts, followed by reduction with ^t^BuOK, gave the tripodal ligands in >85% isolated yields.

### 5.2. Importance of –N(H)– Backbone Functionality in P–C–N Ligands

The –N(H)– functionality can undergo intermolecular N–H^…^O H-bonding to solvent molecules such as MeOH and EtOH. In tripodal ligands such as **4k** intramolecular N–H^…^N H-bonding persists between adjacent arms of the aminomethyl groups on P [74].

The –NH– group was shown to promote protonolysis of L^nacnac^LnR_2_(THF) (Ln = Y and Lu) with 2 equiv. of Ph_2_PCH_2_N(H)Ph yielding phosphinoamido complexes. Further reaction with Ni(COD)_2_ (COD = cycloocta-1,5-diene) resulted in an unusual heterobimetallic species in which one of the P–C bonds was cleaved and the imine group is present within the coordination sphere. Cui et al. [75] showed how [Ph_2_PCH_2_NPh]^−^ chelating amido ligands could be obtained upon deprotonation of the secondary amine, Ph_2_PCH_2_N(H)Ph. Furthermore, unusual heterobimetallic complexes could be obtained in which P–C bond cleavage of one [Ph_2_PCH_2_NPh]^−^ ligand with both the PhN=CH_2_ and PPh_2_ fragments present in the coordination sphere. These findings demonstrate, under these conditions, the instability of the [Ph_2_PCH_2_NPh]^−^ ligand. Johnson and co-workers [68,69,70,71,72,73] used tripodal ligands, based on three amido and one P donor sites, for constructing novel heterometallic complexes. The synthesis of **4j** was accomplished using the water-soluble trialkylphosphine, P(CH_2_OH)_3_ and the desired aniline under neat conditions and with use of dynamic vacuum to remove water. In some cases, it was necessary to use C_7_H_8_ as solvent and Dean-Stark setup to remove water. The ^31^P{^1^H} NMR data for **4j** are shown in Table 1. Reaction of **4j** with excess elemental selenium in C_7_H_8_ gave the corresponding selenides as white solids in >80% isolated yields. The ^31^P{^1^H} NMR spectra showed downfield singlets, flanked with ^77^Se satellites, with J_PSe_ couplings around 700 Hz. Reaction with AlMe_3_ resulted in loss of one of the –CH_2_N(H)Ar arms.

Our group have previously shown that the –NH– group could be further modified to form unsymmetrical PCNCP ligands using a second equiv. of HOCH_2_PR_2_ [76]. These unsymmetrical ligands showed a range of coordination capabilities as a function of the more sterically encumbered R group. More recently, we have also shown the –NH– group could be deprotonated with base, and quenched with ClPPh_2_, to afford PCNP ligands (see Section 8).

## 6. PTA Chemistry

### 6.1. Synthesis of Selected Examples

PTA (1,3,5-triaza-7-phosphaadamantane, Figure 7) is a unique, air stable and water soluble trialkylphosphine that has been extensively studied [77] for its coordination capabilities [78,79,80,81], catalytic [77], and medicinal properties [82]. In 2015, an electron rich tricyclic analogue of PTA, namely CAP (1,4,7-triaza-9-phosphatricyclo[5.3.2.1]tridecane, was reported [77,83,84]. Both PTA and CAP could be prepared by Phospha-Mannich condensations of hexamethylenetetramine or 1,4,7-triazacyclononane and P(CH_2_OH)_3_ [77]. Typically, this procedure involved reaction of commercially available THPC and 1,4,7-triazacyclononane in water/NaOH to give CAP in 39% isolated yield as a white crystalline solid [84]. The ^31^P shows a singlet at 47.8 ppm in CDCl_3_ (Table 1).

Various “upper rim” functionalisations (**5a**, X = various enamines [85], phosphines [86], imidazolyl [87]) have been obtained, via a intermediate lithiated PTA species, thus enabling access to a greater pool of “PTA like” ligand systems. Frost and co-workers [85] showed that lithiated PTA intermediate with aromatic nitriles gave enamine modified ligands in 49–91% yields as white solids. The ^31^P NMR spectra show a typical single resonance around −87.0 ppm. All compounds were shown to slowly oxidise, in the solid state, but in solution this reactivity was more rapid (1 month in chlorinated solvents). Oxidation with H_2_O_2_ afforded the corresponding phosphine oxides. P,N-chelation could be demonstrated by coordination to a W(CO)_4_ fragment. Kwiatkowska and co-workers [88] recently reported the first examples of enantiomerically pure PTA ligands using a series of hydrolytic enzymes in a stereoselective acetylation performed under kinetic resolution conditions.

### 6.2. Importance of -N- Backbone Functionality in PTA and Related Compounds

Whilst various efforts have focused on “upper rim” modification of the carbon atom between P and N, the presence of N donor atoms has enabled alkylations of PTA to be performed using various benzylic halides and all showed good water solubility [89,90]. These alkylated PTA ligands **5b** (A = Cl^−^, Br^−^, PF_6_^−^) could be used as Rh(I) catalyst precursors for the aqueous-biphasic hydroformylation of 1-octene [89] and as Au(I) complexes anticancer agents [90]. Other quaternisations have been reported and used to generate Ru complexes that showed cytotoxic activity towards cancer cell lines [82].

Both PTA and CAP display N-protonation characteristics that were be monitored by ^31^P{^1^H} NMR spectroscopy [77]. Our group found that cationic trialkylphosphines **5c** are intramolecular H-bonded analogues of PTA and could function as effective ligands to Ru(II) and Rh(III) metal centres [91]. PTA was shown to undergo direct N-acylation with benzoic anhydride affording **5d** in 38% yield and was shown to be soluble in both water and other polar solvents [92].

Whilst various examples of complexes of PTA are known, using exclusively the P-donor, it has also been possible to construct networks using both P/N donor atoms [93].

## 7. P–C–N–C–P Chemistry

### 7.1. Synthesis of Selected Examples

As analogues of dppp [bis(1,3-diphenylphosphino)propane], P–C–N–C–P ligands have received widespread appeal. In their excellent review, Balint and co-workers [94] highlighted various aspects of P–C–N–C–P (and P–C–N) ligands and the Reader is directed here for further insights. This type of synthesis methodology (Route B, Figure 1) can be extended to various R/R` groups on both P and N donors and studied, for example in conjunction with Cr, for tri- and tetramerisation of ethylene [95]. In some cases, the ligands **6a** (Figure 8) were found to be extremely air sensitive. Typical R groups on both P centres include Ph and Cy, whilst on N they include ^i^Pr, ^t^Bu, and Ph. The bisphosphine, **6b**, could be prepared under similar conditions as an air stable colourless solid and showed a ^31^P signal at −25.9 ppm [96]. Ligand **6b** was found to react with Au(I) and shown to act as a bridging or chelating diphosphine depending on stoichiometry. In all cases presented so far, the –PR_2_ groups are identical. Our group reported the first examples of nonsymmetrical P–C–N–C–P ligands **6c** using a two-step synthesis [76]. This was further corroborated by the presence of two ^31^P signals in the NMR spectrum consistent with inequivalent P nuclei. We also observed these ligands could bind in various motifs (P-monodentate, PP-chelate, and PP-bridging two different metal centres). The differences in ligating behaviour could be attributed to the sterics of both R groups on the two P-centres. We have also reported the synthesis of a 2-alkenyl N-arene functionalised P–C–N–C–P ligand **6d** [97]. N-pyridyl functionalised bis(phosphino)amines **6e** and **6f** (95% yield, ^31^P −27.7 ppm) could be synthesised from Ph_2_PH, CH_2_O and the appropriate pyridylamine [98,99,100,101]. Both **6e** and **6f** show diverse coordination chemistries with Group 11 metals. Bridging P–C–N (**6g**) [102], 2,2`-bipyridyl diphosphine (**6h**) [103] and the polyphosphine P–C–N–C–P (**6i**) [104,105] ligands have also been reported. Tetradentate ligands **6i**, based on a phenyl, naphthyl or biphenyl scaffold have been prepared and show (by single crystal XRD) weak C–H^…^π interactions upon complexation to Group 11 metal centres [105]. The air stable orange ferrocenyl bisphosphinoamine **6j** could be prepared, in 87% yield, from double condensation of the ferrocenyl primary amine and Ph_2_PCH_2_OH and showed a ^31^P signal at −23.5 ppm [106]. The X-ray structure of this compound was also determined. Likewise the carborane functionalised phosphine **6k** could be prepared from CH_2_O, H_2_NPh in DMF at 60 °C for 3 h and showed two singlets at 30.2 and 36.6 ppm (ratio 30:1 for *rac*:*meso*) for the two diastereomers [107]. Our group recently described a novel diphosphane **6l**, based on two five membered, bicyclic P_2_C_2_N, rings that could be prepared from [P(CH_2_OH)_4_]^+^ and (4-Me_2_N)C_6_H_4_NH_2_ or (4-MeO)C_6_H_4_NH_2_ [108,109]. Microwave assisted Kabachnik-Fields reaction of aminomethylphosphine oxides and (CH_2_O)_n_ and Ph_2_P(O)H gave the corresponding (un)symmetrical phosphine oxides, such as **6m**, in excellent (>90%) yield bearing a central –NH– or –NR– group [110].

Miller and co-workers [111,112,113] prepared tripodal phenyl and cyclopentyl phosphines, **6n**, from Ph_2_P(CH_2_OH)_2_^+^ and NH_3_. The latter in 59% yield and showed a ^31^P signal at −18.4 ppm. Tridentate ligands **6n** can be prepared either from [Ph_2_P(CH_2_OH)_2_]Cl or in situ, from P(C_5_H_9_)_2_H/(CH_2_O)_n_ and NH_3_. The ^31^P{^1^H} NMR spectrum showed a singlet at −18.4 ppm indicative of this substitution pattern. Both P_3_- and P_2_-coordination modes were observed at Ru metal centres. This is a common fragment that is present in a range of phosphine ligands that are finding excellent applications in catalysis and coordination chemistry. In 2011, Gade and co-workers reported the synthesis of tridentate 2,5-dimethyl- and 2,5-diphenylsubstituted phospholanes **6o** using a similar synthetic methodology [114].

### 7.2. Importance of –N(R)– Backbone Functionality in P–C–N–C–P Ligands

Previously, diphosphine **6g** bearing two secondary amine groups, is capable of acting as a bridging ligand. In contrast, the groups of Yamashita [115] and Hill [116,117] have shown related 1,2-substituted bis(phosphino)amines **6p** and **6q** (Figure 9) are precursors to hydroborane and PCP pincer proligands. Hence, reaction of 1,2-phenylenediamine with ^t^Bu_2_PH and CH_2_O gave **6p** in 68% yield, with both –NH– groups available for further reaction, in this case with BH_3_.SMe_2_ and ^n^Pr_2_NH to afford a hydroborane in this case. Whilst **6q** was not isolated, it is clearly an intermediate which subsequently reacts with CH_2_O to form N,N`-bis(phosphinomethyl)-dihydroperimidines.

Our group have been interested, for a number of years, in highly decorated ditertiary phosphines with –CO_2_H and/or –OH functionalities in the N-arene backbone and have found positioning to be important in determining packing arrangements as seen for various Au(I), Pd(II), Pt(II), and Ru(II) metal centres studied [118,119,120,121,122]. The diphosphine **6r** forms an unusual hexameric structure in which the ligand acts as a P_2_O-tridentate ligand [121]. We have shown, by careful manipulation of the R group on the nitrogen atom, the ability to impact a range of packing motifs through H-bonding patterns at various late transition metal centres. In addition, the position of, for example, –CO_2_H groups could also result in intramolecular protonation of one of the P–C bonds forming lactone functionalised P–C–N ligands at a coordinated metal centre [63]. A similar P–C bond cleavage has been observed in a piano-stool complex of **4i** (X = O) which, in the Ru(II) coordination sphere, shows a single secondary aminophosphine ligand and Cp/PPh_3_ ligands [123]. In **6r**, where only a singly *ortho* hydroxy group is available, P_2_O-tripodal coordination at Re(V) and Tc(V) oxo centres has very recently been observed [124].

One of the earliest demonstrations of the importance of the pendant amine is its susceptibility towards protonation, relevant to many catalytic processes involving hydrogen. For example, elegant work by Bullock and co-workers [125] has shown that the tricarbonyl iron complex Fe(CO)_3_(**6s**) undergoes protonation, with [(Et_2_O)_2_H]^+^[B(C_6_F_5_)_4_]^−^ at the Fe, whilst with HBF_4_.OEt_2_, protonation occurs at the iron and pendant N. Treatment with excess HOTf gives a dicationic complex where both the Fe and N centres are protonated. Protonation reactions have also been studied in disubstituted diiron systems as well [126]. In addition to the protonation capabilities at the pendant amine, the nitrogen can also participate in further bonding to a transition metal, acting as a facial P_2_N-tridentate system as found in [Mo(Cp)(PNP-**6s**)(CO)]^+^ [127].

Whilst many studies have focused on the use of Ph_2_PCH_2_OH, the more electron rich Et_2_PCH_2_OH has been used to access a range of amino acid ester ligands **6t** [128]. The Rh complexes have been used for the catalytic hydrogenation of CO_2_ and found to be active with respect to formate formation.

The ligand **6u** has been used to support a Ni(0) metal centre, and immobilised within a protein scaffold via in situ amide bond formation [129]. Finally, Li and co-workers have used a hybrid NHC-diphosphine **6v** as a facial coordinating ligand for the Ru-catalysed synthesis of N-substituted lactams by acceptorless dehydrogenative coupling of diols with primary amines [130].

## 8. P–C–N–P and P–N–N–P Ligands

### Synthesis of Selected Examples

Recent work by our group has shown that rare examples of P–C–N–P ligands **7a** (Figure 10), with an N-backbone group, could be synthesised by reaction of the singly substituted naphthyl P–C–N precursors, with ClPPh_2_ in the presence of LDA [131]. These ligands coordinate to Cr(0) centres generating the corresponding octahedral tetracarbonyl complexes. Furthermore, these ligands were also hown to be effective, in the presence of Cr(acac)_3_ and MMAO-3A, for ethylene tri-/tetramerisations. Conversely, starting from ^t^Bu_2_PNC_3_H_3_N and deprotonation with ^n^BuLi at −78 °C then quenching with ^t^Bu_2_PCl at low temperature then warming to r.t. gave **7b** in 71% isolated yield [132]. Two doublets in the ^31^P NMR support the non-symmetric structure. Only 5% P–N hydrolysis took place in CDCl_3_ indicating **7b** has good solution stability under these conditions. The same approach could be used to access a series of diphosphinoindole ligands **7c** [133]. Reaction of the P(III) intermediate with ^n^BuLi, in Et_2_O, at −78 °C and reaction with ClPPh_2_ gave the chelating ligands **7c** in 35–73% isolated yields. The NMR spectra showed one doublet around 39 ppm for the phosphinoamine and a further second doublet around −25 ppm. In relation to PNP and PCNCP ligands this is a hitherto new class of ligand that has received only limited attention so far.

Simple Ar_2_P–N–N–PAr_2_ ligands, **7d**, can be accessed through the direct reaction of the appropriate hydrazine and ClPR_2_ (Ar = 2-MeC_6_H_4_, 2MeOC_6_H_4_) [48].

## 9. P–C–C–N–C–C–P Ligands

### 9.1. Synthesis of Selected Examples

The basic backbone here represents an important ligand class of terdentate ligand, or “pincer” ligands [134,135] given they can occupy three coordination sites at a metal site. In this context, the central N atom can be viewed as either neutral palindromic or anionic palindromic ligands depending on the charge at nitrogen. Usually the central –NR– group is either a secondary amine, tertiary amine or 2,6-pyridyl group for example. In this case, the role of the central N atom can be more influential on the reactivity of the complex, via electron effects and the variation of the trans influence. Some illustrative examples of ligands of this type are shown in Figure 11 and a brief discussion of the synthesis and reactivity are described here. It is important that these pincer ligands can import good stability and hence often chosen for this property.

We use some recent elegant examples, from the literature, to highlight these classes of diphosphines. As will also be mentioned, these ligands can display noninnocent behaviour and can expand the application of such complexes in transition metal chemistry. This will involve formation of a C=N double bond as will be illustrated and how this can be used to impart further reactivity, either in catalysis and/or through bond activation. Note also, the pincer arrangement also allows outer sphere effects, like what previously seen for the PCNCP ligands with regard to protonation.

The first type of R_2_P–C–C–N(H)–C–C–PR_2_ pincer ligands to consider (Figure 11) are those with a central –N(H)– group where R = Ph (**8a**) [136,137], Cy (**8b**) [138], ^t^Bu (**8c**) [139]. The Ph derivative **8a** was synthesised from Ph_2_PH, ^t^BuOK and (ClCH_2_CH_2_)_2_NH_2_^+^Cl^−^ in THF and could be isolated as a viscous oil showing a ^31^P shift at −20.6 ppm (C_6_D_6_) [136]. The ^t^Bu analogue **8c** was prepared from ^t^Bu_2_PLi and Me_3_SiN(CH_2_CH_2_Cl)_2_ at −60 °C in THF and isolated as a viscous light yellow liquid in 76% yield showing a ^31^P signal at 22.3 ppm (in C_6_D_6_). Tertiary alkylamine diphosphinoamine ligands **8d** [140] and the phenyl-substituted **8e** [141,142] are also known and could easily be obtained, in 44% yield, from reaction of the lithium salt and the bis(chloroethyl)amine HCl salt.

PNP pincer ligands with a pyridyl N atom are also known, both with –CH_2_PR_2_ groups and with –CMe_2_PR_2_ (**8f**) [143,144] or –CH{CH_2_(2-C_5_H_4_N)}PR_2_ (**8g**) [145] substituents. Furthermore, the spacer can also be a –N(H)– group, as opposed to a –CH_2_– group, as is the case for **8h** [146] and prepared by the P–N coupling of ClP(C_6_H_4_CO_2_^t^Bu)_2_ with 2,6-diaminopyridine in the presence of NEt_3_ and showed a ^31^P signal at 25.4 ppm (in CDCl_3_). Unsymmetrical PNP-pincer ligands such as **8i** is an excellent ligand, in conjunction with metals such as Mn [147,148,149,150] or Ru [151] for various catalytic transformations.

Pyridal appended (**8j**) [152], tetradentate P–C–C–N–C–C–P (**8k**) [152] and water-soluble derivatives (**8l**) [153] have also been prepared, the latter via the diprimary phosphine intermediate (H_2_PCH_2_CH_2_)_2_N(CH_2_CH_2_OMe).

### 9.2. Importance of –N(R)– Backbone Functionality in P–C–C–N–C–C–P Ligands

In a recent study in 2022 [154], the synthesis of a large family of N-amide functionalised PNP ligands **8m**, including water-soluble variants [155], has been reported using an acyl chloride, NH_2_(CH_2_CH_2_PPh_2_)_2_^+^Cl^−^ and NEt_3_ in CH_2_Cl_2_. The ^31^P{^1^H} NMR spectra show typically two singlets around −20 ppm, due to restricted amide bond rotation.

Deprotonation of the secondary amine can result in a facial PNP coordination in which there is an amido group. This can be appreciated in several examples (**8n**, **8o**) of compounds shown in Figure 12 [156,157]. Dearomatisation is also important as a function of deprotonation and extensive studies have been undertaken in this field [135].

## 10. Small/Medium Ring Based Cyclic Ligands

### 10.1. Synthesis of Selected Examples

The Mannich condensation reaction can be used to form chiral seven membered macrocycles [158]. As is normal for this synthesis protocol, reaction of the bis(phosphine) with (CH_2_O)_n_ at around 100 °C led to te *rac*/*meso* hydroxymethyl functionalised diphosphine which, upon treatment with chiral amines, gave 1-aza-3,6-diphosphacycloheptanes **9a** as air-stable crystalline solids (Figure 13). The X-ray structure revealed short P^…^P distances (3.120 and 3.118 Å) in comparison to known *rac* isomers. Condensations with arylamines such as aniline, *p*-toluidine and 5 aminoisophthalic acid and benzylamine were undertaken. Two ^31^P peaks at −25 and −27 ppm for the arylamines and −33.5 and −31.8 for benzylamine were observed (Table 1). The reaction of Ph(H)P(CH_2_)_2_P(H)Ph with CH_2_O, afforded the corresponding hydroxymethyldiphosphine, then reaction with ^i^PentNH_2_, in DMF, gave a mixture of products as verified by ^31^P NMR [159]. Isolation of the macrocycle was possible in 21% yield and confirmed by X-ray crystallography. Helm and co-workers [160] expanded this seven-membered ring ligand family to other 4-C_6_H_4_X analogues (X= OMe, H, Me, Br, Cl, CF_3_) and a subsequent series of Ni-based electrocatalysts for hydrogen generation.

Various eight membered cyclo-P_2_N_2_ macrocycles have received attention and been described. For example, cyclic diphosphines P^R^_2_N^R`^_2_ (R = Ph, ^t^Bu; R` = Ph, Bn, **9b**) have been used to prepare diiron [161] and ruthenium [162] complexes. A water-soluble variant of **9b**, bearing –CO_2_H groups, has also been reported by Hey-Hawkins and co-workers [163]. The 1,5-diaza-3,7-diphosphacyclooctane **9c** could be conveniently prepared from (2-C_5_H_4_N)P(CH_2_OH)_2_ and condensation with primary amines leading to a range of air-stable crystalline products. Only one signal was observed in the ^31^P{^1^H} NMR spectra in the region −33 to −63 ppm [164]. Other modifications of the pyridyl cyclic diphosphines [165,166,167] could also be achieved, following similar procedures to that employed for **9c**, and suitably disposed for various coordination studies to be undertaken.

A synthetic strategy based on dynamic covalent chemistry of macrocyclic aminomethylphosphines has been developed for various 14- [168], 16- [169], 18- [170] and 20-membered P_4_N_2_ macrocycles. These reactions start with bis(phosphino)alkanes, formaldehyde and primary amines resulting in multiple products. The amine in question is again also important and influences the outcome of the condensation reaction. Again, the lability of the P–CH_2_–N fragment is important for these dynamic systems.

### 10.2. Importance of –N(R)– Backbone Functionality in Cyclic Diphosphines

As seen in previous sections within this review, the interaction of the pendant amine in complexes of cyclic diphosphines of **9c** (and related analogues) has previously been studied towards protonation reactions [171], heterolytic splitting of H_2_ [172], electrocatalytic alcohol oxidation [173], and hydroalkoxylation [174].

Immobilisation of ligands of the type **9c** have been undertaken in which a suitable para substituted group on the N arene groups has been introduced enabling anchoring to a metal oxide surface (via phosphonic acid groups) [175], glassy carbon electrodes (via a Cu^I^ catalysed alkyne-azide cycloaddition) [176], or a carbon electrode (via amide bond formation) [177]. Recently, Kubiak and co-workers [178] prepared a partially substituted derivative through a multistep approach. The penultimate step, involved lithiation of a bromo intermediate and reaction with ClP(O)(OEt_2_)_2_ in THF at −108 °C. The phosphonate P_2_N_2_ ligand was isolated in 76% yield. A ^31^P spectrum showed a singlet at −49.4 ppm (in CDCl_3_). Here, the group achieved immobilisation via modification of the arene on the P donors (as opposed instead to the N donors as described previously).

## 11. P–C–P–C–N–C–P–C–P and P–C_2_–N–C_2_–N–C_2_–P Ligands

### 11.1. Synthesis of Selected Examples

In this penultimate section, some examples bearing one (or two) amine groups in a diphosphine backbone (Figure 14) are highlighted. Ligand **10a** could be prepared from Ph_2_PCH_2_P(Ph)CH_2_OH and BnNH_2_ as a mixture of meso-/rac- diastereomers [179]. Dissolution in CH_3_CN enabled precipitation of the meso isomer in 25% yield. The ^31^P{^1^H} NMR showed two doublets at −41.7 and −22.3 ppm for the meso form. 1,8-naphthyridine ligands **10b** (R = ^i^Pr, ^t^Bu) [180,181] and 1,10-phenanthroline ligands **10c** (R = Cy, Ph) [182,183] and **10d** [184] are all examples of PNNP-tetradentate ligands. Finally, **10e** could be prepared, as the potassium salt, via reaction of KPPh_2_ with N,N`-bis(2-fluorophenyl)-formamidine in C_7_H_8_ in excellent yield and showed a ^31^P signal at −14.3 ppm (d^8^-THF) [185,186,187]. The 3,3`-azo-benzene phosphine **10f** (^31^P −4.9 ppm, CDCl_3_) was prepared in 66% yield from HPPh_2_, meta-diiodo-azobenzene and Pd(PPh_3_)_4_/NEt_3_ in C_7_H_8_ at 100 °C [188].

### 11.2. Importance of –N(R)– Backbone Functionality

A bisamido, dianionic ligand could be accessed through the neutral parent proligand bisamine **10g** (Figure 15) and deprotonation achieved through reaction with Mg(^n^Bu)_2_/C_7_H_8_ [189] or reaction with a dimesityliron(II) dimer in THF [190]. Lee and Thomas [191] recently found a nickel templated replacement approach of Ph substituents on P could be achieved leading to different –PR_2_ substitutions [with Me, ^i^Pr or –(CH_2_)_3_– groups]. Deprotonation of ligands **10h** [192] and **10i** [193] gave β-diketiminate ligands whose structural flexibility can be realised through complexation to various metal centres.

**Table 1 molecules-27-06293-t001:** ^31^P{^1^H} NMR data for selected ligands discussed in this review.

Ligand	δ(P)/ppm	NMR Solvent	Reference
**1a**	25.9	CDCl_3_	[25]
**1b**	26.4	CDCl_3_	[11]
**1c**	38.5	CDCl_3_	[12]
**1d**	42.2	CDCl_3_	[13]
**1e**	42.1	CDCl_3_	[13]
**1f**	26.1	CDCl_3_	[13]
**1g**	71.3	CDCl_3_	[14]
**1j**	32.6 (and 12.6)	C_6_D_6_	[17]
**1k**	63.9	C_6_D_6_	[18]
**1l**	106.4	CDCl_3_	[19]
**1m**	77.9	C_6_D_6_	[20]
**1n**	126	CD_3_CN	[23]
**dppa**	43.1	CDCl_3_	[31]
**2a**	50.1	CDCl_3_	[32]
**2b**	62.1	CDCl_3_	[35]
**2f**	−6.4	CD_2_Cl_2_	[39]
**2g**	69.0	CDCl_3_	[41]
**2h** ^a^	137.9/135.3	C_7_D_8_	[42]
**2i**	69.3	CDCl_3_	[44]
**2m**	59.5	CDCl_3_	[52]
**2n**	17.2 and −20.0 (*J*_PP_ 277 Hz)	CD_2_Cl_2_	[52]
**3a**	159.8	CDCl_3_	[55]
**3c**	−77.0	CDCl_3_	[56]
**4a**	−16.7	C_6_D_6_	[57]
**4b**	−19.4	C_6_D_6_	[57]
**4c**	−17.1	CDCl_3_	[58]
**4d**	−18.6	CDCl_3_	[59]
**4f**	−26.3	C_6_D_6_	[62]
**4h**	−19.6	C_6_D_6_	[64]
**4i**	*ca*. −61.0	CD_3_COCD_3_	[67]
**4j**	−29.6 to −33.6	C_6_D_6_	[68,69]
**4k**	*ca*. −42.0	CD_3_SOCD_3_	[74]
**PTA**	−98.3	D_2_O	[77]
	−101.0	CDCl_3_	[77]
**CAP**	46.7	D_2_O	[77]
	52.8	CDCl_3_	[77]
	47.8	CDCl_3_	[84]
**5a**	*ca*. −87.0	CDCl_3_	[85]
**5c**	*ca*. −55.0	CD_3_SOCD_3_	[91]
**5d**	−77.9	CDCl_3_	[92]
**6a** ^b^	−26.5	C_6_D_6_	[95]
**6b**	−25.9	CDCl_3_	[96]
**6c**	−27.4 and −41.5 (*J*_PP_ 4 Hz)	CDCl_3_	[76]
**6d**	−27.3	CDCl_3_	[97]
**6e**	−28.0	CD_3_SOCD_3_	[98]
**6f**	−27.7	CDCl_3_	[101]
**6h**	−19.7	CDCl_3_	[103]
**6j**	−25.3	CDCl_3_	[106]
**6k**	30.2 and 36.7	-	[107]
**6l**	*ca*. −34.5	CDCl_3_	[108,109]
**6n** ^c^	−28.0	CDCl_3_	[112]
**6p**	29.5	C_6_D_6_	[115]
**6q** ^c^	−26.0	C_6_D_6_	[116]
**6r**	−22.1 to −28.1	CDCl_3_	[120]
**7a** ^d^	67.0 and −21.7	CDCl_3_	[131]
**7b**	79.6 and 7.0 (*J*_PP_ ~101 Hz)	CDCl_3_	[132]
**7d** ^e^	47.3	CDCl_3_	[48]
**8c**	22.3	C_6_D_6_	[139]
**8d**	*ca*. −19.0	CDCl_3_	[140]
**8e**	−0.4	C_6_D_6_	[141]
**8h** ^f^	25.4	CDCl_3_	[146]
**8k** ^g^	−7.0 and −19.6	CDCl_3_	[152]
**8l** ^h^	−52.8	CDCl_3_	[153]
**8m**	−20.7 and −21.5	CDCl_3_	[154]
**9a** ^c^	−25.8 and −26.6	C_6_D_6_	[158]
**9c** ^i^	−33.5	CDCl_3_	[164]
**10a**	−22.3 and (*J*_PP_ 121 Hz)	CDCl_3_	[177]
**10b**	11.8	C_6_D_6_	[178]
**10e**	−14.3	d^8^-THF	[186]
**10f**	−4.9	CDCl_3_	[188]
**10h**	−14.5	CDCl_3_	[192]

^a^ R = Ph, R` = Me; ^b^ R = Ph, R` = ^i^Pr; ^c^ R = Ph; ^d^ X = H, Y = CH; ^e^ Ar = 2-MeC_6_H_4_^; f^ Ar = 4-CO_2_^t^BuC_6_H_4_; ^g^ R = Cy; ^h^ R = Me; ^i^ R = 2-C_5_H_4_N.

## 12. Catalysis

There has been considerable interest in the development and application of monodentate, bidentate, and polydendate phosphorus containing ligands of various metal complexes as catalysts for wide ranging transformations of academic and industrial relevance. This is also true for phosphine ligands encompassing one (or more) nitrogen donor sites in their ligand backbone structure. The manganese(I) complex MnBr(CO)_3_(P,N-**1b**) was shown to act as a pre-catalyst for the alkylation of amines via reductive amination of aldehydes using molecular H_2_ as reductant [194]. The ruthenium(I) pincer complex RuH(Cl)CO(PN_2_-**1i**) has been shown to hydrogenate catalytically challenging arenols to their corresponding tetrahydronaphthols or cyclohexanols [16,195]. Significant advances have been achieved with chromium catalysts of PNP ligands, for example **2a**, for selective ethylene tri-/tetramerizations [32,33,34]. Furthermore recent examples of note include incorporation of an *N*-triptycene group into the PNP backbone [196], the introduction of bulky -SiR_3_ groups thereby avoiding use of methylaluminoxane [197], and the preparation of new unsymmetrical PNP ligands from Ph_2_PNH(cyclopentyl) [198]. The isopropyl PNP bis(phosphinoamine) ligand **2a** has also been successfully applied to the gold catalysed allylation of aryl boronic acids [199], manganese catalysed dehydrosilylation and hydrosilylation of alkenes [200], and the 2,6-^i^Pr_2_C_6_H_3_ PNP ligand **2f** (and its C_6_H_5_ analogue) were effective in the Buchwald-Hartwig coupling of various sterically hindered substrates [201]. The ready tuneability of PNP bis(phosphinoamine) ligands can be elegantly illustrated by the preparation of PNPO monoxides which can act as P,O-chelating ligands to Pd(II) and Ni(II) to afford catalysts for the copolymerization of ethylene with carbon monoxide [202,203]. Nickel(II) catalysts with cyclic diphosphine ligands incorporating pendant amines have been extensively studied as electrocatalysts for both the oxidation and production of H_2_ [204]. Richeson and co-workers [205] have shown that Ni(II) complexes with the PNP pincer ligands 2,6-{Ph_2_PNR}_2_(NC_5_H_3_) (R = H, Me) can electrocatalytically generate hydrogen from H_2_O/MeCN solutions. Mononuclear iridium(I) complexes of bulky PNNP tetradentate ligands have been shown to be efficient photocatalysts for CO_2_ reduction [206].

## 13. Conclusions

It is without doubt that phosphorus ligands are an important class of compound widely appreciated by the coordination chemistry community. Whilst considerable focus has long been on all carbon backbone P-ligands, there is a considerable growing interest in the incorporation of one (or more) nitrogen atoms. The importance of this class can be released through the facile syntheses of such ligands and the tuneability, in terms of additional reactivities, that can be imparted through the central nitrogen centre. These types of ligands will continue to play a pivotal role in future avenues of phosphorus and transition metal chemistry.

## Data Availability

Not applicable.

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
