# Peer review of "The Backbone of Success of P,N-Hybrid Ligands: Some Recent Developments"

_molecules, 2022, doi:10.3390/molecules27196293_

Round 1
Reviewer 1 Report
This is an excellent review which provides comprehensive and well organised coverage of a topical and important area
Author Response
I thank this Reviewer for their supportive comments on the draft of this review.
Reviewer 2 Report
The review describes a large number of different P,N-hybrid ligands, and the authors mainly focused on the syntheses, structures and their P-NMR characterizations. The manuscript is well arranged, I recommend the publication of the manuscript after some revisions. The main problem of the manuscript is that though the authors said they focused on the coordination chemistry and catalysis of such ligands, however, the catalysis applications of these ligands are less discussed in the review. I think you can add a new paragraph at the end of the review to discuss the catalysis of these ligands. In addition, there are some typing mistakes in the manuscript, an example: page 1, line 28, “Interest has spurned” should be “Interest has spurred”.
Author Response
Following the advice of this Reviewer (and Reviewer 3), we have included a short section (Section 12) and highlight some examples where these ligands have been used in catalysis (thirteen additional references have been included).
I have corrected the spelling mistake picked up here.
Reviewer 3 Report
The manuscript is focused on a synthetic chemistry of bi- and polydentate phosphine ligands containing amino moiety in the ligand skeleton as well as some coordination chemistry and catalytic applications. It is very well written and may be accepted for publication after minor revision.
1) The resolution in all Figures is disappointingly low with apparent pixelization. It will be also useful to draw double C=O bond in ligand 4g (Figure 6) to avoid any ambiguity.
2) Many recent catalytic applications of complexes bearing these ligands are omitted. I perfectly understand that it is difficult to consider the whole literature on this topic, however I recommend to cite some more recent examples with remarkable results: ACS Catal. 2017, 7, 4446; Chem. Commun., 2018, 54, 4302; ChemCatChem 2019, 11, 4351; Dalton Trans. 2019, 48, 2730; J. Catal. 2021, 404, 163; Chem. Eur. J. 2021, 27, 13518; J. Am. Chem. Soc. 2021, 143, 10743; Nat. Chem. 2021, 13, 182.
Author Response
1) All the figures, originally imported as .png files, have now been replaced with .tif files to address the issue of resolution quality. Ligand 4g now includes a C=O double bond.
2) The Reviewer is thanked here for their comment and indeed, given the vast nature of this field, it has been impossible to capture all significant examples in this field. We have included a new section as previously mentioned above.